# Oncogenic Signalling Pathways in Cancer Immunotherapy: Leader or Follower in This Delicate Dance?

**DOI:** 10.3390/ijms26094393

**Published:** 2025-05-06

**Authors:** Douglas Cartwright, Andrew C. Kidd, Sonam Ansel, Maria Libera Ascierto, Pavlina Spiliopoulou

**Affiliations:** 1School of Cancer Sciences, University of Glasgow, Bearsden, Glasgow G61 1QH, UK; douglas.cartwright@glasgow.ac.uk (D.C.); andrew.kidd@glasgow.ac.uk (A.C.K.); sonam.ansel@glasgow.ac.uk (S.A.); maria.ascierto@glasgow.ac.uk (M.L.A.); 2Beatson West of Scotland Cancer Centre,1053 Great Western Road, Glasgow G12 0YN, UK

**Keywords:** immunotherapy, oncogenic drivers, targeted therapy, resistance mechanisms

## Abstract

Immune checkpoint inhibitors have become a mainstay of treatment in many solid organ malignancies. Alongside this has been the rapid development in the identification and targeting of oncogenic drivers. The presence of alterations in oncogenic drivers not only predicts response to target therapy but can modulate the immune microenvironment and influence response to immunotherapy. Combining immune checkpoint inhibitors with targeted agents is an attractive therapeutic option but overlapping toxicity profiles may limit the clinical use of some combinations. In addition, there is growing evidence of shared resistance mechanisms that alter the response to immunotherapy when it is used after targeted therapy. Understanding this complex interaction between oncogenic drivers, targeted therapy and response to immune checkpoint inhibitors is vital for selecting the right treatment, at the right time for the right patient. In this review, we summarise the preclinical and clinical evidence of the influence of four common oncogenic alterations on immune checkpoint inhibitor response, combination therapies, and the presence of shared resistance mechanisms. We highlight the common resistance mechanisms and the need for more randomised trials investigating both combination and sequential therapy.

## 1. Introduction

Following the introduction of the anti-cytotoxic T-lymphocyte associated protein 4 (CTLA-4) inhibitor, ipilimumab, in 2011, immunotherapy in the form of immune checkpoint inhibitors (ICIs) has transformed the treatment of solid cancers. The key benefit of immunotherapy over cytotoxic chemotherapy or targeted therapy is the long-term responses seen well after completion of therapy, even in patients with advanced disease [1]. As a result, the use of ICI has expanded across multiple tumour types, both as monotherapy and in combination with alternative ICIs, cytotoxic chemotherapy or targeted therapies. For example, the anti-programmed cell death protein 1 (PD-1) inhibitor, pembrolizumab, has licenced indications in 18 different cancer types alone [2].

Alongside the development of ICIs, the recognition and targeting of common oncogenic driver mutations are changing the landscape of cancer therapeutics. Improvements in the understanding of kinase function and design of small molecule inhibitors have led to the development of targeted therapy blocking aberrant signalling due to mutations that were previously considered “undruggable” [3]. There is growing evidence that the presence of alterations in oncogenic drivers not only predicts response to targeted agents but influences response or resistance to other therapies, such as ICIs [4]. A major effect of the activation of many oncogenic pathways is immune modulation and the generation of an immunosuppressive microenvironment. As a result, combining targeted therapies with ICI has become an attractive therapeutic option. However, overlapping toxicity profiles have limited the clinical use of some of these combinations [5].

## 2. Mechanisms of Immune Checkpoint Inhibitor Resistance

To explore the mechanisms of ICI resistance, we need to understand how an immune response develops. An effective immune response requires components of both innate and adaptive immunity. The innate immune response involves the recognition of cancer cells by dendritic cells through both neoantigen-dependent and independent mechanisms [6]. Mature dendritic cells migrate to lymphoid tissue and engage the adaptive immune response [6]. In addition, dendritic cells release pro-inflammatory cytokines, resulting in the infiltration of immune cells and cancer cell death through macrophage and natural killer cell-mediated mechanisms. Although the research on ICI resistance has focused on the T cell response, there is increasing recognition of the importance of effective innate immunity on ICI response [7].

The adaptive immune response is triggered by the presentation of tumour neoantigens by dendritic cells and other antigen-presenting cells within lymphoid tissues. These bind to the T cell receptor (TCR) on naïve T lymphocytes. Full activation of T lymphocytes occurs with the addition of costimulatory signals, such as CD28–CD80/CD86 [8]. Activated tumour-specific T lymphocytes undergo clonal expansion and are trafficked to the tumour. Tumour-specific cytotoxic CD8^+^ T cells recognise tumour cells through the presentation of neoantigens displayed by human leukocyte antigen (HLA) proteins. Cytotoxic CD8^+^ T cells trigger tumour cell death through the release of cytolytic molecules such as granzyme A/B, perforin and cathepsin C [9]. A subset of activated T lymphocytes differentiates into memory T cells and produces an effective response on rechallenge of the antigen. However, in addition to costimulatory pathways that amplify T cell activation, inhibitory pathways, such as CTLA-4/CD80, programmed death ligand1 (PD-L1)/PD-1 and lymphocyte-activation gene 3 (LAG-3) prevent excessive immune response. Upregulation of these inhibitory pathways is one way that tumours evade immune surveillance. ICIs block these inhibitory pathways leading to amplification of T cell activation and an effective immune response.

Even in highly immunogenic tumours such as melanoma, where long-term response rates can be as high as 60%, ~25% of patients have primary ICI resistance [1]. This is defined as progression within 6 months of starting ICI treatment [10]. The remaining patients who do not experience long-term response develop acquired resistance or progression after an initial response of >6 months [10]. Several factors influence the initial response to ICI. Genomic alterations such as tumour mutational burden (TMB), microsatellite instability (MSI) or mismatch repair (MMR) deficiencies are associated with improved response to ICI [11,12,13]. These genomic alterations result in neoantigen formation leading to T-cell activation. In addition, the expression of immune checkpoints, such as PD-L1, and the pattern of T cell infiltration are also associated with improved response to ICI [14,15,16]. Many of the underlying mechanisms of primary and acquired resistance are shared. Several reviews have categorised these mechanisms, including into tumour cell intrinsic and extrinsic mechanisms [17,18,19,20,21].

### 2.1. Tumour Cell Intrinsic Mechanisms

Intrinsic factors affecting ICI response are defined as alterations in specific pathways within tumour cells that lead to resistance [20]. A diminished number of neoantigens is the key mechanism of primary resistance [17]. This can be a result of the reduced production of neoantigen, such as low TMB, or through deficient antigen presentation mechanisms. Alterations in β-2-microglobulin (*B2M*) have been found in clinical samples from melanoma patients treated with ICI [22,23], and this finding has also been validated in other pan-cancer studies. *B2M* is a component of major histocompatibility complex (MHC) class I proteins and is involved in antigen presentation. Loss of heterozygosity is enriched in non-responders compared to responders [23,24].

Genomic alterations can also affect immune cell trafficking and infiltration. Tumour cell WNT/β-catenin signalling results in T cell exclusion due to the reduced expression of chemokine ligands 4 (CCL4) and impaired recruitment of CD103^+^ dendritic cells, which are involved in T cell activation [25]. Phosphatase and tensin homologue (PTEN) loss, which leads to increased phosphoinositide 3-kinases (PI3K) signalling, reduces T cell infiltration through the expression of immunosuppressive cytokines such as vascular endothelial growth factor (VEGF) [26]. Upregulation of the mitogen-activated protein kinase (MAPK) pathway also results in increased PI3K signalling and expression [27]. Mutations in the Janus kinase (JAK) signal transducer and activator of the transcription (STAT) pathway lead to a reduction in the response to interferon-γ signalling, which is critical to antigen presentation and immune cell recruitment [28,29].

### 2.2. Tumour Cell Extrinsic Mechanisms

Alongside the tumour cell intrinsic factors, environmental factors and supporting cells within the tumour microenvironment (TME) play a key role in resistance to immunotherapy. Regulatory T cells (Tregs) suppress the immune response through the secretion of immune suppressive cytokines and direct contact with effector T cells [30,31]. However, the role of Tregs in ICI resistance is complex. Infiltration of Tregs, and specifically the ratio of CD8^+^/Treg, is associated with immunosuppressive TME and poor prognosis in several cancers [32,33]. CTLA-4 is constitutively expressed on Tregs and is, therefore, a key target of anti-CTLA-4 inhibitors such as ipilimumab. Within murine models, the response to anti-CTLA-4 inhibitors may be associated with depletion of Tregs and reversal of the CD8^+^/Treg ratio [34]. In contrast, within human tumours, anti-CTLA-4 inhibitors do not deplete Treg cells but they abrogate Treg suppressive function [35]. In addition, high baseline FOXP3^+^ Tregs infiltration may be associated with clinical response to anti-CTLA-4 inhibition [36]. However, in patients with hyper-progression after anti-PD-1 therapy, an increase in proliferative PD-1^+^ Tregs within the tumour and circulation has been observed. In summary, Tregs infiltration of Tregs is a potential mechanism of ICI resistance but it is likely to be tumour and Treg subtype-specific.

Myeloid-derived suppressor cells (MDSCs) are immature cells of myeloid lineage which have an immunosuppressive effect within the TME. They are derived from the bone marrow and consist of monocytic (CD14^+^CD15^−^HLA-DR^lo/–^) or polynuclear (CD11b^+^CD14^−^CD15^+^/CD66b^+^) subtypes [37]. Multiple trials have shown that the presence of circulating MDSCs is associated with poor response to immune checkpoint inhibition [38,39,40]. The mechanisms that underly the function of MDSCs within cancer are broad and summarised by Lasser S et al. [41]. Briefly, they can promote the infiltration of Tregs and interact with tumour-associated macrophages (TAMs) to promote an immunosuppressive TME [42,43]. In addition, MDSCs express high levels of ARG1 that metabolise l-arginine to urea and ornithine. L-arginine is required for T cell activation and proliferation; therefore, depletion of l-arginine by MDSC-derived ARG1 can reduce T cell activation [44,45]. MDSCs also contribute to oxidative stress within the TME, which impairs T cell effector functions [46,47].

Similar to MDSCs, TAMs are an abundant myeloid cell type within the TME. TAMs can be defined as either antitumour (M1) or pro-tumorigenic (M2) subtypes; however, this reflects a continuum between the two states rather than distinct subtypes [48]. TAMs suppress T cell function through expression of PD-L1 [49]. Cytokines secreted by TAMs, such as TGF-β1 and chemokine (C-C motif) ligand 22 (CCL22), stimulate and recruit Tregs to the TME [50,51]. In addition, TAMs can also affect ICI pharmacokinetics and lead to resistance through binding of the Fc domain of the antibody to their Fcγ receptor [52].

The physical properties of the TME also influence the immune response. The extracellular matrix (ECM) is a collagen-rich scaffold predominantly produced by cancer-associated fibroblasts (CAFs) in the TME. Increased ECM stiffness reduces T-cell migration and response to ICI [53]. In addition to the effect on Tregs, TGFβ stimulates the formation of CAFs which leads to the production of a desmoplastic stroma and T-cell exclusion [54]. Preventing CAF differentiation through inhibition of NOX4 can overcome the resistance to anti-PD-1 therapy in murine models [55].

Although resistance mechanisms can be classified into tumour cell intrinsic and extrinsic mechanisms, they do not exist in isolation. Indeed, there is significant crosstalk between mechanisms that together promote an immunosuppressive TME. For example, activation of PI3K or MAPK pathways results in the altered expression of inflammatory cytokines, such as VEGF, that can influence immune cell infiltration and CAF activation [26,56,57]. CAFs in turn, secrete C-X-C motif chemokine 5 (*CXCL5*), which binds to c-x-c motif chemokine receptor 2 (CXCR2) on cancer cells, resulting in PI3K pathway activation and downregulation of PD-L1 [58]. This complex crosstalk between tumour cell intrinsic and extrinsic resistance mechanisms suggests that inhibition of a single pathway will be insufficient to overcome immunotherapy resistance.

In summary, the mechanisms that lead to ICI resistance are diverse; connecting a variety of intrinsic cell pathways to components of the tumour microenvironment, which ultimately leads to immunosuppression.

## 3. Oncogenic Drivers

### 3.1. RAS/RAF/MAPK Pathway

Components of the mitogen-activated protein kinase (MAPK) pathway are frequently altered in human cancer. Approximately 20% harbour a Rat sarcoma virus (*RAS*) mutation, 7% harbour *BRAF* mutations and <1% harbour mitogen-activated protein kinase kinase 1 (*MEK*) mutations [59,60]. *RAS* proteins are GTPases, which act as a primary switch that can activate several downstream signalling cascades, including RAF/MAPK/ERK and PI3K pathways. Out of the three *RAS* isoforms (*KRAS*, *NRAS*, *HRAS*), *KRAS* is the most commonly mutated in approximately 95% of pancreatic cancers, 50% of colorectal cancers and 30% of lung cancers [61,62,63]. The majority are found in codons 12 or 13 and specific point mutations are associated with different tumour types. *KRAS G12C* mutations are commonly found in lung adenocarcinomas, and *G12D* mutations are more commonly found in pancreatic and colorectal cancers. *NRAS* mutations are present in approximately 11% of all cancers and are commonly found in 15% of melanomas [64,65]. *HRAS* mutations are much rarer, affecting only 4% of cancers [64].

RAF kinases are a collection of serine/threonine kinases that transduce the signalling cascade from *RAS* towards *MEK* protein. Activating mutations in BRAF are the most common alterations and are found in approximately 50% of metastatic melanoma [53]. Over 90% of these occur in codon 600 [66,67]. *BRAF* mutations are separated into three classes [68]. Class I mutation confers *BRAF* activity as monomers and this class includes *BRAF V600E*. Class II mutations require dimerisation for constitutive *BRAF* activity [69]. Finally, class III mutations result in impaired kinase function and require alternative *RAS* activation for activity [69].

#### 3.1.1. Effect of RAS/RAF/MAPK Pathway Alteration on Immunotherapy Response

The effect of the *RAS*/*RAF*/*MAPK* pathway on tumour immune response has been extensively researched. *KRAS* mutation results in the higher expression of C-X-C motif chemokine receptor 2 (*CXCR2*) ligands, which recruits immunosuppressive neutrophils and monocytes [56]. KRAS mutations also increase CD47 expression, which is a myeloid checkpoint preventing phagocytosis by antitumour macrophages [70]. One downstream effect of MAPK signalling is the stabilisation of *PD-L1* mRNA resulting in increased expression of PD-L1 [71,72]. However, *KRAS* and *BRAF* mutation may have contrasting effects on immune infiltration. Within colorectal cancer, *BRAF* mutant tumours have higher levels of immune infiltration compared to *BRAF* wild type [73]. In contrast, *KRAS* mutant tumours had lower immune cell infiltration compared to *KRAS* wild type. This observation may be a result of a high proportion of *BRAF* mutations in microsatellite instable (MSI-high) colorectal cancers, particularly in the older, non-hereditary MSI-high cancers [74]. In addition, it may represent the differences in the alternative pathways activated in *KRAS* mutant cancers, such as *PI3K*, compared to the *BRAF* mutant MAPK pathway activation.

Clinically, the presence of *KRAS* or *BRAF* mutations on ICI response has not been fully established and varies across tumour types. Within advanced unresected melanoma, *NRAS* or *BRAF* mutation has little effect on ICI response. Combination ipilimumab and nivolumab resulted in progression-free survival (PFS) of 11.2 months in patients with *BRAF* mutant melanoma compared to 11.7 months in *BRAF* wild-type tumours [1]. No difference in PFS and overall survival (OS) to monotherapy or combination immunotherapy was found between *NRAS* mutant and wild type in a prospective multicentre study [75]. However, a meta-analysis assessing the overall response rate (ORR) in *NRAS* mutant and *NRAS* wild-type patients indicated that NRAS mutant tumours have a better response rate to ICI [76]. This study only looked at ORR and there was evidence of the disproportionate influence of a single study on the pooled analysis.

Within lung cancer, the retrospective IMMUNOTARGET registry showed that *KRAS* mutations were associated with improved response to ICI compared to other oncogenic mutations [4]. However, due to differences in PD-L1 testing across centres, the authors were not able to determine if this was due to the higher PD-L1 expression previously reported in *KRAS* mutant tumours [71]. In addition, the study did not include a control cohort without the presence of targetable oncogenic mutations; therefore, it is not possible to compare the response rate to the whole non-small cell lung cancer (NSCLC) population and previous trials.

ICIs are limited to MSI-H tumours in colorectal cancers. However, there is a higher incidence of sporadic MSI in *BRAF* mutant colorectal cancer. The presence of *BRAF* or *KRAS* mutation had no significant effect on PFS and OS in the Keynote 177 trial of pembrolizumab compared to chemotherapy in MSI-H advanced colorectal cancer [77,78].

#### 3.1.2. Combination Immune Checkpoint Inhibition with RAS/RAF/MAPK Targeted Therapy

Preclinical studies have shown a clear rationale for combining targeted therapy for *RAS* and *RAF* mutations with ICI. *BRAF* inhibition (BRAFi) with or without *MEK* inhibitors increased neoantigen expression and CD8^+^ T cell infiltration and reduced immunosuppressive cytokine production [27,79]. Clinical trials of combination BRAF/MEKi and ICI have shown mixed results. Keynote022 in patients with melanoma showed an improved PFS of triplet pembrolizumab with dabrafenib and trametinib in comparison to dabrafenib and trametinib alone (median PFS was 16.9 months vs. 10.7 months) [80]. However, the COMBI-I trial of triplet spartalizumab with dabrafenib and trametinib showed no benefit in PFS or ORR [5]. In addition, the rate of grade 3–5 toxicities with triplet therapy is high and has prevented combination therapy from being adopted into routine clinical practice.

Preclinical studies have shown that *KRAS* inhibitors produce a pro-inflammatory microenvironment and have synergistic effects in combination immunotherapy. Sotorasib, a *KRAS G12C* inhibitor, increased CD8^+^ T cell infiltration and the recruitment of CD103^+^ dendritic cells required for T cell priming [81]. In combination with anti-PD-1 therapy, sotorasib significantly improved the response rate and duration of response in the CT-26 KRAS^G12C^ colorectal murine model, compared to sotorasib or anti-PD-1 monotherapy [81].

The codebreak 100/101 trial is assessing the safety and efficacy of combination sotorasib and either pembrolizumab or atezolizumab in advanced *KRAS G12C* mutant NSCLC [82]. The response rates across all cohorts are approximately 30%; however, this includes patients previously treated with ICI and all dose escalation cohorts where the patient may have received subtherapeutic doses. However, similar to combination BRAF/MEKi and immunotherapy, there are significant rates of G3-4 hepatotoxicity that may limit clinical use. The *KRAS G12C* inhibitor, MK-1084 is explored in phase 3 clinical trials in combination with pembrolizumab [83]. Phase 1 trial data shows a promising response rate of 47%; however, this was in untreated patients [84]. A combination of targeted therapy and ICI could be a valuable therapeutic strategy if the toxicity profile can be improved. Specifically, it may delay the onset of resistance to targeted therapy.

#### 3.1.3. Shared Resistance Mechanisms with RAS/RAF/MAPK Targeted Therapy

BRAF/MEK inhibitors are the first targeted therapy to show a clear effect on response to subsequent ICI therapy [85,86]. The exclusion of effector T cells is a key feature of shared resistance, which can result from both tumour cell intrinsic and extrinsic resistance mechanisms. Tumour cell intrinsic resistance mechanisms include the activation of alternative oncogenic signalling pathways, such as increased *WNT5A* expression leading to WNT/β-catenin upregulation [87]. WNT signalling is associated with T-cell exclusion and an immunosuppressive environment [25]. In addition, the upregulation of PI3K signalling leads to reduced effector T-cell recruitment through increased VEGF secretion [26]. Tumour cell extrinsic resistance mechanisms include the promotion of CAF formation and ECM remodelling resulting from increased TGF-β expression BRAFi-resistant melanoma cells [88]. BRAFi resistance is also associated with infiltration of immunosuppressive TAMs, which support tumour growth and immune escape [89].

The clinical effect of these shared resistance mechanisms was demonstrated in the DREAMSeq and SECOMBIT trials assessing the sequencing of BRAF/ MEKi and ICI in metastatic melanoma [85,86]. In both trials, BRAF/MEK inhibition until progression followed by ICI resulted in worse OS compared to an ICI-first therapeutic strategy. However, upfront immunotherapy was inferior to BRAF/MEKi in the first year of treatment, which is likely due to the longer time-to-response seen with ICI. Short-term BRAF/MEKi followed by ICI (‘sandwich approach’) may preserve responsiveness before resistance emerges [85].

Clinical data on the effect of resistance to KRAS inhibitors on response to subsequent immunotherapy is not yet available. However, several studies have shown similar resistance mechanisms to BRAF/MEKi that may influence the immune microenvironment. Mutation in the downstream components of KRAS, such as BRAF, MEK and PI3K, results in the activation of alternative signalling and KRAS inhibitor resistance and potentially reduces T cell recruitment [26,90,91]. In addition to the gain in function mutations, loss of function mutations in PTEN have also been found in clinical samples taken at the progression of disease during KRAS inhibition [92]. PTEN loss is associated with immunotherapy resistance through activation of the PI3K/AKT pathway and alterations in the antiviral interferon network [93]. KRAS inhibitors can also stimulate aerobic glycolysis, leading to an acidic tumour microenvironment through the secretion of lactate [94]. This can inhibit the function of immune cells within the tumour microenvironment.

#### 3.1.4. Summary of RAS/RAF/MAPK Pathway

Alterations in the RAS/RAF/MAPK pathway contribute to an immunosuppressive tumour microenvironment; however, there is no clear association with poor response to ICI. Preclinical studies and early trial data have shown a rationale for combined therapy that may prolong responses, particularly to KRAS inhibitors. However, overlapping toxicity profiles may limit combination approaches with targeted therapies such as BRAF/MEK inhibitors. Emerging resistance mechanisms to targeted treatment may influence response to subsequent ICI, particularly in BRAF mutant melanoma. If combination therapy is not possible, an ICI-first approach may be preferable to elicit a durable tumour regression, preserving the targeted treatment for the timepoint of immunotherapy resistance when swift rescue may be required.

### 3.2. FGFR Pathway

The fibroblast growth factor receptor (FGFR) is a receptor tyrosine kinase family of four transmembrane receptors [95]. Aberrations in these receptors are detected in 5–10% of human cancers; however, there is a higher frequency in urothelial cancer and intrahepatic cholangiocarcinoma (32% and 15%, respectively) [96,97].

#### 3.2.1. Effect of FGFR Alterations on Immunotherapy Response

Fibroblast growth factor (FGF)/FGFR signalling pathways are involved in the development and differentiation of cells, as well as in angiogenesis and carcinogenesis [95]. The downstream effects of FGF pathway activation have a diverse impact on immunotherapy response. Potential FGFR signalling-induced tumour cell intrinsic ICI resistance mechanisms include deficiencies in antigen presentation. *FGFR* signalling can influence neoantigen presentation via MAPK pathway-mediated inhibition of MHC I and II expression, as well as reducing B2M expression through inhibition of INF-γ stimulated *JAK*/*STAT* signalling [98,99]. *FGFR1* signalling led to the increased expression of PD-L1 and resistance to anti-PD-1 therapy in the LL2 lung carcinoma mouse model, via MAPK pathway activation [100]. In addition, *FGFR1*-mediated YAP upregulation initiates PD-L1 transcription in squamous cell lung cancer [101]. *FGF2* amplification rather than *FGFR1* amplification may be the predominant mechanism for increased FGFR1 signalling as it correlated with higher PD-L1 expression in human lung squamous cell carcinoma and urothelial cancers [101,102]. However, not all FGF receptors have the same effect. Activating mutations in FGFR3 are associated with the reduced expression of PD-L1 in urothelial cancers in a real-world patient cohort [103]. FGFR3 has been shown to activate the E3 ubiquitin ligase, NEDD4, which polyubiquitinates PD-L1, leading to its degradation [104].

In addition to contributing to tumour cell intrinsic ICI resistance, FGF/FGFR pathway alterations may also be involved in tumour cell extrinsic ICI resistance. Analysis of a FGFR-mutant genetically engineered mouse model (GEMMs) of lung cancer revealed a TME with low T cell infiltration [105]. Increases in CD8^+^, CD4^+^ T cells and M1 macrophage infiltration have been demonstrated in FGFR1 knockout models [101]. This is consistent with clinical data on triple-negative breast cancer, where FGFR expression is associated with decreased CD8^+^ T cell infiltration and increases in immunosuppressive M2 macrophages [106]. Upon FGFR1 activation in mouse mammary epithelial cells, macrophage recruitment has been demonstrated via the induction of CX3CL1, an inflammatory chemokine [107]. However, the effect of FGFR1 expression may not be consistent across tumour types. The expression of FGFR1 positively correlates with CD8^+^, CD4^+^ and macrophages in human gastric cancer [108].

The clinical data of what effect alterations in the FGF/FGFR pathway have on response to ICI is largely from retrospective cohort data and small series rather than prospective trials. A study examining four ICI-treatment studies concluded that FGFR-mutant melanoma conferred a better objective response rate compared to patients with FGFR-wild-type melanoma [109]. The incidence of FGFR mutations in this cohort was 22% (*n* = 119/529), with FGFR2 being the most prevalent mutation (10%). In contrast, a study of ICI-treated patients that included melanoma, lung, colorectal and breast cancer patients from the TCGA database, identified that amplifications in the FGFR ligand, FGF2, are associated with immunotherapy resistance [110]. This is in agreement with a small cohort of four patients with hyper-progression, where amplifications in FGF2, FGF4 or FGF19 were identified in three patients (75%) [111]. One possible mechanism for this is interferon-γ driven activation of the FGF2—pyruvate kinase M2 (PKM2)—β catenin pathway, which is associated with hyper-progression following ICI [112]. Finally, immunotherapy response in urothelial cancers is poorest in the luminal-papillary subtypes [113]. Luminal-papillary tumours are enriched in FGFR3 alterations and are associated with T cell deplete phenotype [114,115,116].

#### 3.2.2. Combination Immune Checkpoint Inhibition with FGFR Inhibitors

There are currently three FDA-approved selective small-molecule inhibitors of FGFR: erdafitinib, futibatinib, and pemigatinib [117,118,119]. In addition, there are several multi-targeted TKIs that variably inhibit FGF receptors. These include sorafenib, lenvatinib, sunitinib, and pazopanib [120]. These multi-target TKIs may affect ICI response through several pathways; therefore, deciphering the specific role of FGFR inhibition in this setting is challenging. However, the preclinical literature supports the use of both selective and multi-target TKI FGFRi-ICI combination therapy, with a particular emphasis on the enhancement of the immune-depleted TME in FGFR-mutated cancers [105,115,121]. FGFR blockade using erdafitinib in a mouse model of triple-negative breast cancer led to increased T cell infiltration via MAPK/ERK downregulation [106]. Anti-PD-1 therapy in combination with lenvatinib yielded improved T cell function in the tumour microenvironment in HCC tissue samples subjected to RNA-sequencing [121]. Increased CD4^+^ helper and CD8^+^ effector T cell infiltration, reduced T_regs_ and downregulated PD-L1 expression on cancer cells have been reported in *FGFR2*- and p53-mutated lung cancer mouse models treated with erdafitinib in combination with anti-PD-1 therapy [105].

Clinical data for selective FGFR inhibitors in combination with ICIs is currently immature, with only preliminary results from phase I/II trials available. The phase II Norse study investigated combination erdafitinib and an anti-PD-1 monoclonal antibody, cetrelimab, with erdafitinib monotherapy in FGFR-altered urothelial cancer [122]. ORR in the combination was higher in the combination arm compared to monotherapy (54.5% vs. 44.2%) and higher 12-month OS (68% vs. 56%) [122]. For multi-target TKI drugs, combination therapies with ICI are already in clinical use. Lenvatinib is approved in combination with pembrolizumab in renal cell cancer and endometrial cancer [123,124]. The dual treatment rationale is two-fold: the anti-angiogenic effects of combined VEGFR1–3 and FGFR1–4 inhibition, and cytotoxic T cell-mediated increase in IFN-γ [125,126,127]. In the Keynote-775 trial of advanced endometrial cancer, lenvatinib in combination with pembrolizumab was compared to investigator’s choice of chemotherapy [124]. Combination therapy resulted in a longer median PFS (6.6 months vs. 3.8 months) and higher ORR (31.9% vs. 14.7%) compared to chemotherapy. However, due to the comparator arm being chemotherapy, the benefit of combination therapy compared to monotherapy ICI or lenvatinib is unclear.

The CLEAR-trial in advanced renal cancer compared lenvatinib plus pembrolizumab to lenvatinib plus everolimus or sunitinib monotherapy [123]. The trial showed a longer median PFS in the lenvatinib plus pembrolizumab (23.9 months, 95% CI 20.8–27.7 months) compared to lenvatinib plus everolimus (14.7 months, 95% CI 11.1–16.7 months). However, the trial was designed to compare to sunitinib monotherapy and not identify differences in lenvatinib combinations [123].

In the LEAP-002 trial in advanced HCC, lenvantinib in combination with pembrolizumab was compared to lenvatinib with placebo [128]. However, the trial failed to meet the primary endpoints of significant difference in OS and PFS, suggesting no additional benefit of pembrolizumab to standard lenvatinib monotherapy in this setting. Importantly, the addition of pembrolizumab to lenvatinib and everolimus did not significantly increase toxicity, with similar incidence of grade 3–4 toxicities to lenvatinib monotherapy, lenvatinib plus everolimus, or sunitinib [123,128]. The combination Lenvatinib plus Pembrolizumab had higher rates of grade 3–4 toxicities than chemotherapy in Keynote-755 (88.9% vs. 72.7%) [124].

#### 3.2.3. Shared Resistance Mechanisms with FGFR Inhibitors

A common resistance mechanism to FGFR inhibition is mutations in the FGFR kinase domain, which limits the access of the TKI to the hydrophobic binding pocket [129]. However, retrospective analysis of a small series of patients with longitudinal ctDNA testing revealed new alterations in the MAPK pathway as an additional emerging resistance mechanism to FGFR inhibitors [130]. As discussed above, MAPK pathway activation can contribute to ICI resistance. In vitro studies of FGFRi-resistant lung cancer cell lines confirmed MAPK activation as a potential biomarker of resistance and also identified the overexpression of AXL [131]. AXL is a part of the TAM receptor family and can activate a variety of downstream signalling pathways, including RAS/RAF/MAPK. AXL signalling may contribute to ICI resistance as observations indicate that AXL overexpression is associated with an immunosuppressive microenvironment [132]. In addition, AXL inhibition improves response to ICI in preclinical models [133].

Clinical evidence of sequential therapy is limited to the use of TKIs after the progression on ICI. Erdafitinib has been used in combination with ICI, yielding encouraging objective rates in the phase II BLC2001 trial with subsequent FDA approval [117]. Twenty-two patients (total cohort n = 99) had previously received ICI. Within this subgroup, 13 (59%) achieved a response to erdafitinib. Interestingly, only one patient in this subgroup responded to their prior course of ICI. This is consistent with phase II trial data, discussed above, which indicates that FGFR alterations in urothelial cancer are associated with poor response to ICI [113]. In addition, these data suggest that prior ICI does not alter the response to subsequent FGFR inhibition.

Phase 2 LEAP-004 trial assessed the combination of pembrolizumab and multi-target TKI, lenvatinib, in patients with melanoma who were ICI-resistant [134]. Most patients had primary resistance to ICI (70.9%), indicating an extremely poor prognosis cohort. ORR to combination lenvatinib and pembrolizumab was 28.2%, as determined by iRECIST, and a median duration of response (DOR) of 12 months. Patients who received the combination treatments were afforded a median PFS of 4.2 months and an OS of 14 months. These data indicate that combination multi-target TKI and ICI may be a valuable treatment strategy in ICI resistance; however, the specific role of FGFR inhibition in the mechanisms is not clear.

Although there is preclinical evidence to suggest a potential shared resistance mechanism that would affect ICI response after progression on FGFR targeted therapy, to the best of our knowledge, there have been no clinical trials assessing response rates to this form of sequential therapy.

#### 3.2.4. Summary of FGFR Pathway

Alterations in the FGF/FGFR pathway are associated with an immunosuppressive microenvironment; however, the effect on ICI response may be tumour-type specific. The blockade of FGFR-activated pathways may mediate remodelling of the T-cell-depleted phenotype (110,111), thus enhancing the co-action of immune checkpoint inhibition. Combination therapy with lenvatinib and pembrolizumab has demonstrated clinical activity and is entering clinical practice, with further trials of specific FGFRi and ICI currently ongoing (see Table 1). Early evidence suggests ICIs do not significantly impact subsequent FGFRi response, consistent with the findings with BRAF/MEKi in BRAF mutant melanoma. Further studies are required to determine if the FGFRi-first strategy may influence long-term outcomes.

### 3.3. PI3K

Phosphoinositide 3-kinases (PI3K) are an enzymatic group of three classes [135]. The alpha isoform (p110α) of phosphatidylinositol 3-kinase (PI3K) is encoded by the *PIK3CA* gene and is the most frequently mutated in human cancers [136,137]. PI3K has a role in cell growth, migration and apoptosis, as well as carcinogenesis [138,139,140]. PI3K overactivation results in elevated AKT, an oncogenic signalling protein. Abnormal cell cycle progression follows PI3K/AKT constitutively active signalling [141]. AKT activation also leads to phosphorylation of mTOR and subsequent cancer progression through signals of proliferation [135].

Aberrant PI3K signalling can be attributed to somatic loss of PTEN [142]. PTEN loss results in elevated phosphatidylinositol-3,4,5-trisphosphate (PtdIns(3,4,5)P3) and upregulation of the PI3K-AKT-mammalian target of the rapamycin (mTOR) pathway [143]. Additionally, PI3K point mutations have been identified in solid organ cancers [144]. *PIK3CA* mutations are common in solid organ cancers [144]. Breast cancer (12.5–41.1%), endometrial cancer (20–34%), colorectal cancer (13–18%) and NSCLC (3.7–19%) have the highest prevalence of PI3K mutations [145,146,147,148,149,150].

PI3K mutational status confers a poorer prognosis in metastatic breast cancer [151]. It is also predictive of PI3K inhibitor response in patients with oestrogen receptor-positive breast cancer and head and neck squamous cell cancers [152,153]. A similarly negative effect on prognosis is observed in PI3K-mutated endometrial cancer [154]. However, in colorectal cancer and NSCLC, PI3K is not associated with adverse survival [155,156].

#### 3.3.1. Effect of PI3K Mutations on Immune Checkpoint Inhibitor Response

As discussed above, PI3K activation is a recognised mechanism of acquired resistance to ICI [26]. PI3K signalling is central to several tumour cell intrinsic ICI resistance mechanisms due to crosstalk between MAPK, NF-κB and Wnt/β-catenin pathways [135]. PI3K pathway activation alters the secretion of cytokines, such as VEGF and CCL2, thus promoting an immunosuppressive TME [26,157].

Pre-existing activating mutations in PIK3 can also lead to an immunosuppressive microenvironment through tumour cell intrinsic mechanisms. For example, mRNA and PD-L1 protein levels are associated with PI3K mutations in cervical cancer [158]. Of the 250 patients studied, those who were PD-L1 positive were more likely to have a *PIK3CA* mutation (79% vs. 53.4%, *p* = 0.019). Similarly, PI3K/AKT signalling has also been shown to promote PD-L1 expression in breast cancer [159]. The authors examined the effect of recombinant high-mobility group box 1 (rHMB1) through receptors for advanced glycation end products (RAGE) on cell migration in human breast cancer cell lines, concluding that after down-regulating AKT, the subsequent PI3K inhibition dephosphorylated AKT, and prevented PD-L1 expression [159].

In addition to tumour cell intrinsic mechanisms, PI3K mutations can lead to immunotherapy resistance through tumour cell extrinsic mechanisms. *PIK3CA-H1047R* mutations, a potent driver of carcinogenesis, result in a reduction in CD8^+^ T cell infiltration and an increase in myelosuppressive myeloid populations [160,161]. This leads to an attenuated response to anti-PD-1 therapy, which could be rescued by the pharmacological inhibition of PI3K [161]. Moreover, patients with wild-type PIK3CA had a greater percentage of CD8^+^ in tissues compared with those with a PIK3CA-E545K mutation [158]. The upregulation of PI3K-mTOR signalling has been shown to inhibit the activation and differentiation of T cells [162].

In contrast to the immunosuppressive effects of PI3K pathway activation, PI3K mutations can be associated with higher tumour mutational burden and immune cell infiltration in certain cancer types [153,163]. The analysis of patients with head and neck squamous cell carcinoma treated with immunotherapy from the MSKCC-2019 cohort found that PI3K pathway mutations were associated with longer OS [153]. The mechanisms behind this contrasting effect of PI3K mutations have not been identified.

#### 3.3.2. Combination Immune Checkpoint Inhibitors with PI3K Inhibition

PI3K pathway influences immune checkpoint expression [164]. As such, combination treatment with ICIs and PI3K/AKT/mTOR inhibitors has attracted research interest. In a BRAF mutant, PTEN-null mouse model of melanoma, combination inhibition with anti-PI3Kβ inhibitor and anti-PD-1 antibody improved response and survival compared to either monotherapy [26]. Combination therapy significantly increased the infiltration of CD4^+^ and CD8^+^ T cells [26]. This is consistent with the novel pan-PI3K inhibitor, KTC1101, which also results in increased CD4^+^ and CD8^+^ T cell infiltration and improved response to anti-PD-1 therapy [165]. KTC1101 acts directly on immune cells to modulate T cell populations but also stimulates cancer cells to produce inflammatory cytokines, such as chemokine ligand 5 (*CCL5*) and C-X-C motif chemokine ligand 10 (CXCL10), which promotes CD8^+^ T cell chemotaxis [165]. Dual PI3K and immune checkpoint inhibition has been shown to decrease T_regs_ and optimise memory CD8^+^ T cells in immunotherapy resistance models in vivo and in vitro [166]. The pan-class I PI3K inhibitor, buparlisib, in combination with anti-PD-1 therapy, significantly inhibited tumour growth in the PyMT mammary tumour murine model compared to buparlisib or anti-PD-1 monotherapy alone [167].

Alpelisib, a PI3K inhibitor that targets the p110α subunit, is currently the only FDA-approved PI3K inhibitor in advanced breast cancer [168]. Clinical evidence of combination alpelisib with ICI is lacking. However, in a two-patient case series of PIK3CA mutated metastatic squamous cell cancer of head and neck cancer, the addition of alpelisib to pembrolizumab or nivolumab resulted in sustained clinical response [169]. The limitation of this series is that treatment was not conducted within a clinical trial, which raises doubts about the validity of the data and lack of standardised response assessment. A phase II trial of the pan-PI3K inhibitor, copanlisib, showed response rates of up to 27% in combination with nivolumab, even in patients pretreated with ICIs [170]. This study was a biomarker-driven study selecting patients with PIK3CA mutations or PTEN loss. In an unselected population, combination copanlisib and nivolumab has an ORR of 18% in a phase Ia study [171]. Several studies are currently investigating combination PI3K inhibition with ICI (summarised in Table 1).

#### 3.3.3. Shared Resistance Mechanisms with PI3K Inhibitors

The predominant mechanism for PI3K inhibitor resistance is the reactivation of the PI3K/AKT pathway either through *PI3K* mutation, its upstream or downstream signalling components, or positive feedback mechanisms [172]. As a result of crosstalk between numerous oncogenic pathways, as described above, activation of the MAPK pathway has also been identified in PI3K inhibitor-resistant PDX models of triple-negative breast cancer [173]. However, to the best of our knowledge, there are no clinical studies investigating the immunological effect of PI3K inhibitor resistance or its influence on ICI response.

#### 3.3.4. Summary of PI3K Targeted Therapy

PI3K mutations have a potential role as a predictive biomarker of immunotherapy response. Preclinical evidence provides a rationale for combining PI3K inhibitors with ICI, providing an exciting option for patients with PI3K pathway mutations or *PTEN* loss. However, clinical data on the safety and effectiveness of this strategy is immature. Resistance mechanisms, such as reactivation of the PI3K/AKT pathway, or activation of the MAPK pathway, may influence response to subsequent immunotherapy.

### 3.4. HER2/ERBB2 Alterations

The v-erb-b2 avian erythroblastic leukaemia viral oncogene homolog 2 (*ERBB2*) gene, also known as the human epidermal growth factor receptor 2 (*HER2*) gene, encodes a member of the epidermal growth factor receptor family of receptor tyrosine kinases. *HER2* alterations can be broadly classified as amplification, overexpression and mutations [174]. *HER2* gene amplification and the resultant overexpression of *HER2* protein is associated with cell transformation and oncogenesis, and *HER2*-directed therapy has demonstrated success in some *HER2* overexpressing (i.e., *HER2*-positive) cancer types such as breast and gastro-oesophageal cancers. Around 20% of breast cancers have *HER2* gene amplification, with 15–20% overexpressing the *HER2* protein [175]. In 1998, Trastuzumab was the first *HER2* targeted treatment to be approved for *HER2*-positive metastatic breast cancer and since then its use has expanded to treat all stages of breast cancer and many novel anti-HER2 therapies have followed [174].

Oncogenic *HER2* activation through somatic gene mutation has been studied and the majority of these *HER2* mutations are not linked to *HER2* gene amplification [176]. Mutations are found across all exons of the HER2 gene and mutations affecting the extracellular domain, transmembrane domain, or tyrosine kinase domain may activate *HER2* signalling pathways, which promote cell cycle progression and proliferation. Exon 20 insertions (ex20ins) affecting the kinase domain are the most common *HER2* mutations. The Cancer Genome Atlas (TCGA) dataset shows that mutations occur in a variety of cancers; bladder cancer has the highest prevalence of *HER2* mutations (9–18%), followed by uterine cervix (6%), colorectal (5.8%), lung (4%) and breast (4%) [177]. Pahuja et al. analysed over 100,000 tumours and detected HER2 mutations in 3.5% of them [178].

#### 3.4.1. Effect of HER2 Alterations on Immunotherapy Response

Recent data suggest that *HER2* status (i.e., the amount of HER2 protein in *HER2* overexpressing tumours) may influence the tumour immune microenvironment, and response to *HER2*-targeted therapies can be predicted by immune cell compositions [179]. Tumour infiltrating lymphocytes (TILs) provide antitumour immunity, and higher levels of TILs are associated with improved distant disease-free survival in breast cancer patients following Trastuzumab treatment [180]. It is thought that PD-L1 is associated with HER2-status and agents targeting the HER2 pathway may relieve inhibition of antitumour immunity. Koung Jin Suh et al. used gastric cancer cell lines and resected gastric tumour samples to study the effect of *EGFR/HER2* signal blockade on the tumour immune microenvironment of tumours overexpressing *HER2* [181]. The authors concluded that inhibition of the *EGFR*/*HER2* signalling pathway suppressed PD-L1 and released immunosuppressive cytokines, suggesting that *EGFR/HER2* inhibition may create a more favourable milieu for tumour immunotherapy [181]. On the contrary, somatic *HER2* mutations in solid tumours may improve the tumour microenvironment to favour immunotherapy. Wang et al. analysed patient data from eight studies to investigate the effects of *HER2* mutations on immune checkpoint inhibitor outcomes [176]. Objective response rates were higher in those with HER2 mutations compared to those with *HER2* wild type (44.4% vs. 25.7%, *p* = 0.081) and patients carrying the mutation also had better overall survival [176].

*HER2* alterations are present in 7–27% of de novo NSCLC and may serve as a resistance mechanism in up to 10% of *EGFR*-mutated NSCLC [182]. Activating *HER2* mutations are rare in NSCLC, with a prevalence of 2–4%, but they are successfully targeted with trastuzumab or trastuzumab deruxtecan (T-DXd) [183]. T-DXd is approved as a second-line treatment for patients with *HER2*-mutant NSCLC. Several cohort studies have found that NSCLC patients with *HER2* ex20ins have higher tumour mutational burden but low PD-L1 expression compared to those with *HER2* non-ex20ins [183,184,185]. Furthermore, a recent meta-analysis found that patients with non-ex20ins had better progression-free survival [13.0 vs. 3.6 months] and overall survival (27.5 vs. 8.1 months) following ICI compared to ex20ins patients, consistent with findings of the META-ICI cohort [183]. The authors also reviewed immune characteristics and found that the TME was generally immunosuppressed in *HER2*-mutated NSCLC [183]. Interestingly, immune signatures revealed that non-ex20ins patients might be enriched with resting CD4^+^ memory T cells, which requires further investigation in a larger cohort [183]. Furthermore, the IMMUNOTARGET registry, which identified HER-2 alterations in 5% (n = 27) of patients with NSCLC, also linked HER-2 perturbations to immunotherapy resistance, with a poor objective response rate to ICI monotherapy (7%) [4]. Due to the low incidence of HER2 alterations in NSCLC, all the cohorts in these studies are small and this hinders the validity of the findings. A larger prospective study of NSCLC with HER-2 aberrations would add more statistical power to the true predictive effect of HER2 alterations on ICI response or resistance.

#### 3.4.2. Combination Immune Checkpoint Inhibition with HER2 Targeted Therapy

Preclinical data has shown promising results supporting the use of ICI in combination with *HER2*-directed therapies in *HER2*-positive breast cancer. Trastuzumab emtansine (T-DM1) in combination with anti-PD-1 antibody and anti-CTLA-4 antibody has been shown to improve the efficacy of ICI in mouse models by synergistically activating CD8^+^ T cells [186]. Anti-PD-1 therapy in combination with trastuzumab has also shown improved antitumour effects in preclinical models [187]. T-DXd promotes an antitumour immune response through an increase in MHC I expression and dendritic cell markers [188]. In addition, combination T-DXd with anti-PD-1 improved response rates in an immunocompetent CT26.WT-hHER2 mouse model [188].

The success in preclinical studies has resulted in several clinical trials of combination therapies. The PANACEA trial tested the combination of pembrolizumab (anti-PD-1) and trastuzumab in pretreated HER2-positive metastatic breast cancer patients [189]. The combination was effective for patients with tumours resistant to trastuzumab-based therapies and positive for PD-L1 biomarker, with a partial ORR of 15%, but not in patients with PD-L1-negative tumours [189]. This led to the DIAmOND clinical trial, which is currently investigating the use of dual immunotherapy (durvalumab and tremelimumab) with trastuzumab in patients with advanced HER2-positive breast cancers [190]. Atezolizumab, an anti-PD-L1 antibody, is being tested in combination with T-DM1 (NCT04740918) in patients with trastuzumab-resistant PD-L1-positive HER2-positive advanced breast cancer; the results of this trial are awaited. Avelumab, another anti-PD-L1 antibody is being trialled in combination with trastuzumab and vinorelbine (NCT03414658) in progressing HER2-positive breast cancer.

Despite the promising preclinical evidence, the benefit of adding ICI to anti-HER2 therapy may not be universal for all HER2-altered cancers. The Keynote-811 trial assessed the addition of pembrolizumab to trastuzumab and chemotherapy in locally advanced or metastatic HER2-positive gastric or gastro-oesophageal cancer, [191]. At the third interim analysis, there was a trend towards improved PFS and OS with the addition of pembrolizumab compared to placebo, but this did not reach statistical significance. The combination was tolerable with only a small increase in the incidence of grade 3 or higher adverse effects (58% vs. 51%) [191].

#### 3.4.3. Shared Resistance Mechanisms with HER2 Targeted Therapy

Similarly to resistance mechanisms emerging after other targeted therapies, activation of downstream signalling, or activation of an alternative oncogenic pathway associated with ICI resistance, have been identified in treatment resistance following anti-HER2 therapy [192]. Activation of PI3K and PTEN loss is associated with trastuzumab resistance [193,194].

In addition to the PANACEA trial described above, the KATE2 trial assessed the role of the ICI, atezolizumab, in HER2-positive breast cancer following disease progression on trastuzumab [195]. The combination of atezolizumab with trastuzumab emtansine demonstrated a trend towards longer PFS compared to trastuzumab emtansine and placebo (8·2 months vs. 6·8 months) but did not reach statistical significance (*p* = 0.33). In addition, the objective response rates were similar between groups (45% in atezolizumab arm, 43% in the placebo arm) [195]. Interestingly, PD-L1 status, as assessed by *PD-L1* gene expression but not IHC, was associated with improved response to atezolizumab in subgroup analysis. Both the results of PANACEA and KATE2 suggest that although responses are possible after progression on anti-HER2 therapy, ICI is not sufficient to overcome anti-HER2 resistance, and PD-L1 status may be a useful biomarker.

#### 3.4.4. Summary of HER2 Targeted Therapy

The presence of HER2 alterations is associated with improved response to immunotherapy in breast cancer. Preclinical data supports the use of combination HER2-directed therapy and ICI. However, clinical trials of combination therapy have not shown statistical benefit in all cancer types and all patients. PD-L1 status may be a useful biomarker for combination therapies, particularly in breast cancer. HER2-alterations in NSCLC are rare and although T-DXd is licensed for HER2 mutant NSCLC, further studies are required to assess it in combination with ICI. Mechanisms of resistance to *HER2*-directed therapy share the hallmarks of other targeted therapies that may influence response to ICI. These include the upregulation of alternative pathways, such as PI3K pathway activation. The lack of benefit with the addition of atezolizumab to trastuzumab emtansine in patients resistant to trastuzumab may indicate the resistance to ICI after *HER2*-directed therapy. However, further trials are required to investigate this.

## 4. Discussion

As sequencing technology improves, tumour genomic profiling will become more common in routine clinical practice. The challenge for clinicians and researchers is to understand how to utilise this information to personalise treatment approaches for individual patients. As summarised in this review, the presence of oncogenic driver alterations influences the immune microenvironment and response to immunotherapy (see Figure 1). However, the effects are complex and vary across tumour types. Combining ICI with targeted therapy has the potential to overcome the early acquired resistance often seen with targeted therapies and improve immune response. Balancing improved efficacy with high rates of grade 3/4 toxicity will be challenging.

Overlapping toxicity profiles has been a major factor that has hindered the introduction of combination therapies with the incidences of grade 3 toxicities as high as 82% [196]. Alternative approaches, such as lead-in dosing strategies where monotherapy-targeted agents are administered prior to combination therapy, may reduce high toxicity rates [82]. However, further work is needed to determine the effect on long-term toxicities and efficacy.

In many tumour types, targeted therapy and ICI will be used sequentially. BRAF mutant metastatic melanoma is the only setting where we have dedicated randomised trials to assess the sequencing of targeted therapy and ICI. This is in part due to the early introduction of ICI in metastatic melanoma resulting in long-term clinical data. In addition, BRAF inhibitors and ICI were both approved by the FDA for metastatic melanoma in 2011. Therefore, both treatments were valid first-line treatment options for BRAF mutant metastatic melanoma. The SECOMBIT and DREAMseq trials clearly show that shared resistance mechanisms influence response, and the choice of sequential therapy is important [85,86]. Many of the resistance mechanisms discussed in this review are shared across different targeted therapies (summarised in Table 2). This may suggest that the clinical benefit of ICI before targeted therapy, seen in BRAF mutant melanoma, may be replicated in other tumour types or oncogenic drivers. However, it would be speculation to define the optimal sequencing of therapy in other pathways or cancer types at present due to the lack of specific clinical trial evidence. Nevertheless, we need to consider the potential for shared resistance when introducing new therapies and determine where they may be most effective from the outset. Therefore, we propose a framework for how this clinical question may be approached as new evidence emerges (Figure 2).

### 4.1. Future Perspectives and Challenges

Biomarkers that either predict the resistance or identify the emergence of resistance are going to be a key part of precision oncology. New technologies, such as monitoring of circulating tumour DNA (ctDNA), can identify the development of new mutations that confer resistance to targeted therapies [197]. These may also help to identify activating mutations in pathways associated with ICI resistance. ctDNA analysis provides a practical and clinically acceptable method for serial molecular profiling throughout a treatment course. However, not all tumours shed ctDNA and nearly a quarter of genomic alterations detected by tissue methods are not identified on ctDNA analysis [198]. Alterations with low variant allele frequency (VAF) suggest low clonality [199]. VAF can be predictive of response to therapy; however, due to the variations in assays, there are no agreed standards for the assessment of VAF or the clinical significance of low VAF alterations.

Tissue samples provide a more direct assessment of the oncogenic drivers present within a tumour and are easier to interpret. Next-generation sequencing (NGS) has been widely adopted in clinical trial design and is beginning to be adopted into routine clinical practice. However, the turnaround time for the NGS of tumour samples remains approximately 30 days even in well-resourced tertiary centres [200]. This timeframe is too long for some patients with rapidly progressing diseases to wait for treatment decisions. Sequencing costs are falling but the interpretation of NGS results remains complex and often requires the input of an expert molecular tumour board. This limits its utility in resource-poor settings. Therefore, a broad approach not requiring tumour profiling and targeting both intrinsic and extrinsic mechanisms may provide the best clinical utility.

Tissue-based NGS and ctDNA can predominantly identify tumour intrinsic ICI resistance mechanisms through the detection of new mutations. As a result, the majority of the shared resistance mechanisms discussed above are tumour cell intrinsic. To detect tumour cell extrinsic mechanisms, spatial profiling is required to characterise the infiltration of specific immune cell populations and the composition of the TME. Spatial technologies, such as multiplex immunofluorescence, spatial transcriptomics, proteomics and metabolomics, have been used in preclinical research to identify the mechanisms of resistance to ICI [201,202]. However, these technologies are currently not clinically applicable. Currently, spatial technologies are costly and often low throughput, limiting their use to small cohorts [203]. Tissue microarrays are often used to overcome this limitation but are inherently flawed due to intra-tumour heterogeneity.

In this review, we have not examined the role of novel immunotherapies as the data on these therapies, particularly in relation to specific driver mutations, is limited. Novel immunotherapies, such as TIL cell therapy, are effective treatment strategies for ICI resistance [204]. However, these treatments are currently limited by high cost, complex manufacturing processes and long lag times for producing individual patient treatments, which limits their widespread adoption.

### 4.2. Conclusions

The presence of oncogenic drivers has a significant impact on the TME and response to ICI. Many targeted therapies have a synergistic effect with ICI and can improve response rates if tolerable combinations are found. In therapies where combination therapy is ineffective or intolerable, shared resistance mechanisms can be influential. To maximise tumour response and disease control from each line of therapy, large clinical trials focusing on the sequencing of treatment options may be proven valuable.

## Figures and Tables

**Figure 1 ijms-26-04393-f001:**
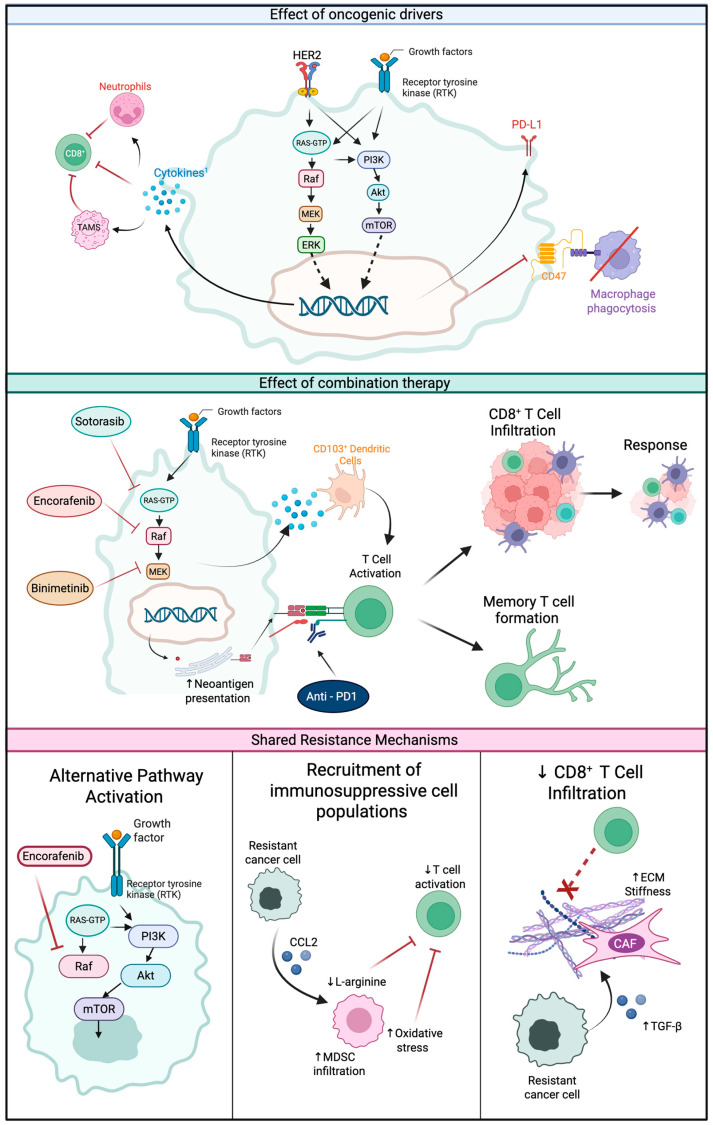
**Effect of oncogenic drivers.** An overview of the interaction between oncogenic drivers on immune microenvironment. Amplification in growth factors, such as members of the fibroblast-like growth factor family (FGF), or mutations in receptor tyrosine kinases trigger activation of downstream signalling pathways. These include the RAS/RAF/MAPK pathway and PI3K pathways. Activation of these pathways induces expression of immunosuppressive cytokines, such as CXCR2 and VEGF, that inhibit CD8^+^ T cell recruitment and activation. In addition, RAS/RAF/MAPK and PI3K pathways regulate the expression of checkpoint proteins such as PD-L1 and CD47. **Effect of combination therapy.** A summary of the rationale of combining targeted therapy with immunotherapy using inhibition of the RAS/RAF/MAPK pathway as an example. Inhibition of the RAS/RAF/MAPK pathway leads to a reduction in immunosuppressive cytokines and recruitment of CD103^+^ dendritic cells required for CD8^+^ T cell activation. In addition, BRAFi increases neoantigen expression. Combining these effects with ICI results in T-cell activation, resulting in an immune response and tumour regression. In addition, the formation of memory T cells promotes durable responses. **Shared resistance mechanisms.** A summary of the three types of shared resistance mechanisms. Inhibition of one pathway, in this example BRAF inhibition with encorafenib, results in upregulation of alternative pathways such as PI3K. Altered expression of key cytokines, such as CCL2, results in the recruitment of immunosuppressive cell populations, including myeloid-derived suppressor cells (MDSCs). MDSCs can suppress T cell function by a reduction in metabolites required for activation, such as L-arginine, and increase oxidative stress. Other cytokines, such as transforming growth factor-β (TGFB), promote cancer-associated fibroblast (CAF) activation, which inhibits T cell infiltration through increased extracellular matrix (ECM) stiffness. Created in BioRender. Cartwright, D. (2025) https://BioRender.com/i02o781 (accessed on 18 March 2025).

**Figure 2 ijms-26-04393-f002:**
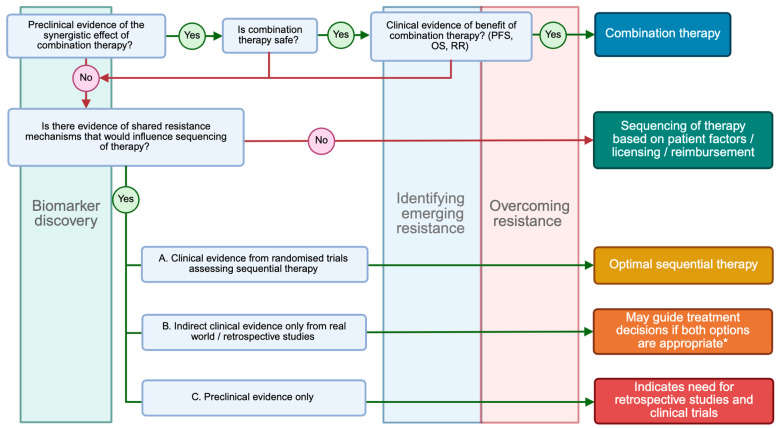
Framework for determining the optimal utilisation of ICI and targeted therapy. For combination therapy there needs to be evidence of a synergistic mechanism of action, the combination is safe and tolerable, and an objective benefit in randomised clinical trials. The measure used to assess the benefit of combination therapy could be improved response rate (RR), overall survival (OS), or progression-free survival (PFS), but may include other endpoints that might be relevant in specific clinical scenarios. If combination therapy is not appropriate, then we must look for evidence of shared resistance mechanisms. If none exist then the choice of sequencing of therapy is determined by patient factors (e.g. comorbidities, concomitant medications etc.), practical implications of delivering treatment, and the licensing and reimbursement of the treatment in the individual healthcare service. If there is evidence of shared resistance mechanisms, then a randomised trial of sequential therapy is required to determine the optimal sequential therapy. In the absence of randomised clinical trials, retrospective cohort studies and real-world data may influence the sequential therapy treatment decision if both treatments were appropriate options for the patient (* considering licencing, approvals and patient factors). Treatment decisions cannot be based purely on preclinical evidence; however, it can inform the priorities for retrospective studies, clinical trial design and translation research. Alongside all these questions are the overarching priorities to identify biomarkers that can select which patients may benefit most, how we identify the emerging resistance in patients, and how we can overcome it.

**Table 1 ijms-26-04393-t001:** Summary of ongoing clinical trials investigating immune checkpoint and targeted therapy combinations. Trials were identified through clinicaltrials.gov (accessed on 10 January 2025).

Names	Target	Targeted Agent	Immune Checkpoint Inhibitor	Phase	Tumour Type	Status
RAS/RAF/MAPK Pathway						
NCT03600883	KRAS G12C	Sotorasib	Pembrolizumab/Atezolizumab	I/II	NSCLC	Active, not recruiting
NCT04185883	KRAS G12C	Sororasib	Pembrolizumab/Atezolizumab	IB	NS	Recruiting
NCT05920356	KRAS G12C	Sotorasib	Durvalumab (PD-L1)	II	NSCLC	Recruiting
NCT04613596	KRAS G12C	Adagrasib	Pembrolizumab/Atezolizumab	II/III	NSCLC	Recruiting
NCT05789082	KRAS G12C	Divarasib	Pembrolizumab	IB/II	NSCLC	Recruiting
NCT04449874	KRAS G12C	Divarasib	Atezolizumab	IA/IB	NS	Recruiting
NCT03235245	BRAF/MEK	Encorafenib/Binimetinib	Ipilumumab/Nivolumab	II	Melanoma	Active, not recruiting
NCT04238624	BRAF/MEK	Dabrafenib/Trametinib	Cemiplimab	I	Thyroid	Active, not recruiting
NCT04061980	BRAF/MEK	Encorafenib/Binimetinib	Nivolumab	II	Thyroid	Active, not recruiting
NCT05926960	BRAK/MEK	Encorafenib/Binimetinib	Pembrolizumab/Ipilumab/Nivolumab	II	Melanoma	Active, not recruiting
FGFR Pathway						
NCT06511648	FGFR	Erdafitinib	Cetrelimab	II	Bladder	Recruiting
NCT05036681	FGFR	Futibatinib	Pembrolizumab	II	Endometrial	Recruiting
NCT04601857	FGFR	Futibatinib	Pembrolizumab	II	Urothelial	Active, not recruiting
NCT04828486	FGFR	Futibatinib	Pembrolizumab	II	HCC	Active, not recruiting
NCT05945823	FGFR	Futibatinib	Pembrolizumab	II	NS	Recruiting
NCT06263153	FGFR	Futibatinib	Durvalumab (PD-L1)	II	Bladder	Recruiting
NCT05004974	FGFR	Pemigatinib	Sintilimab	II	NSCLC	Recruiting
NCT06389799	FGFR	Pemigatinib	Retifanlimab	II	Liposarcoma	Recruiting
PI3K Pathway						
NCT06545682	PI3K	Alpelsib	Pembrolizumab	IB	Breast	Recruiting
NCT04975958	PI3K	Buparlisib	Atezolizumab	IA	NS	Active, not recruiting
HER2/ERBB2 Alterations						
NCT04740918	HER2	Trastuzumab emtansine	Atezolizumab	III	Breast	Terminated
NCT03414658	HER2	Trastuzumab emtansine	Atezolizumab	III	Breast	Active, not recruiting
NCT04448886	HER2	Sacituzumab Govitecan	Pembrolizumab	II	Breast	Active, not recruiting
NCT03747120	HER2	Trastuzumab/Pertuzumab	Pembrolizumab	II	Breast	Active, not recruiting
NCT03125928	HER2	Trastuzumab/Pertuzumab	Atezolizumab	II	Breast	Active, not recruiting
NCT04759248	HER2	Trastuzumab	Atezolizumab	II	Breast	Recruiting
NCT03417544	HER2	Trastuzumab	Atezolizumab	II	Breast	Active, not recruiting
NCT04873362	HER2	Trastuzumab emtansine	Atezolizumab	III	Breast	Active, not recruiting

Not specified (NS), non-small cell lung cancer (NSCLC), hepatocellular carcinoma (HCC).

**Table 2 ijms-26-04393-t002:** Summary of interactions between targeted therapy and ICI exploring synergistic mechanisms of combination therapy, tumour cell intrinsic mechanisms and tumour cell extrinsic mechanisms. MHC = major histocompatibility complex, Tregs = regulatory T cells, CAF = cancer-associated fibroblast, TAM = tumour-associated macrophage, TME = tumour microenvironment. Light blue colour indicates there is evidence of synergistic mechanisms of action of the targeted therapy and ICI. Dark blue indicates that there is evidence that resistance to the targeted therapy results in tumour cell intrinsic ICI resistance. Light green colour indicates that there is evidence of tumour cell extrinsic ICI resistance.

	Targeted Therapy
	KRAS	BRAF	MEK	FGFR	PI3K	HER2
Synergistic mechanism of action with ICI					
Increased CD8+ T cell infiltration						
Increased CD4+ T cell infiltration						
Dendritic cell infiltration						
Increased neoantigen expression						
Increase MHC class I expression						
Downregulation of Tregs						
Downregulation of PD-L1						
Reduced Immunosuppressive cytokine secretion						
Intrinsic resistance mechanisms					
PTEN loss/PI3K activation						
Alternative MAPK pathway activation						
WNT/β Catenin signalling						
Extrinsic resistance mechanisms					
CAF activation						
TAM infiltration						
Acidic TME through aerobic glycolysis

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
