# Peer review of "Oncogenic Signalling Pathways in Cancer Immunotherapy: Leader or Follower in This Delicate Dance?"

_ijms, 2025, doi:10.3390/ijms26094393_

Round 1
Reviewer 1 Report
Comments and Suggestions for Authors
The review „Oncogenic signalling pathways in cancer immunotherapy: leader or follower in this delicate dance?” focuses on presenting the preclinical and clinical evidence showing how oncogenic alterations affect the response to immune checkpoint inhibitors, a combination of targeted therapy and ICI, and the presence of shared resistance mechanisms.
The manuscript is timely and has the potential to be of interest to experts in the field. In the context of tumor immunology and immunotherapy, the subject matter is both pertinent and novel. This manuscript is well-organized and detailed, with a logical flow. The references are appropriate. I did not find any inaccuracies or striking omissions in the literature. The Table is appropriate. A figure aids comprehension of processes that are discussed in the text. I have several minor comments to be addressed:
1) A separate section could be included by authors, such as 'Conclusions and Future Perspective' or something similar.
2) Line 708: Section 2.2.” Figures, Tables and Schemes” does not contain any Figures, Tables or Schemes. Shouldn't that headline be removed?
Author Response
- A separate section could be included by authors, such as 'Conclusions and Future Perspective' or something similar.
We have updated the discussion to include a future perspective and conclusion section. (Line 688 onwards).
- Line 708: Section 2.2.” Figures, Tables and Schemes” does not contain any Figures, Tables or Schemes. Shouldn't that headline be removed?
We have resized the figure to fit onto the same page as the section heading.
Reviewer 2 Report
Comments and Suggestions for Authors
Summary
This review paper thoroughly investigates the complicated interplay between oncogenic signaling pathways (RAS/RAF/MAPK, FGFR, PI3K, and HER2) and immune checkpoint inhibitors (ICIs) in cancer therapy. It assesses the impact of oncogenic changes on immune evasion, resistance mechanisms, and the efficacy of combined or sequential ICI-targeted therapy. The authors combine preclinical and clinical data to highlight potential and limitations in tailoring immunotherapy for genetically defined cancer populations.
Strengths
- The discussion includes both tumor-intrinsic and -extrinsic resistance mechanisms and spans across several key pathways and cancer types.
- The review including many recent publications (2021–2024), which supports the currency of the review.
Recommendations:
- Please define abbreviation (e.g., TME) at first usage.
- Some languages are verbose: “A sandwich approach for short-term BRAF/MEKi…” could be more precise: “Short-term BRAF/MEKi followed by ICI (‘sandwich approach’) may preserve responsiveness before resistance emerges.”
- The review paper is dense. For readability, please breaking down long paragraphs in sections 3.1-3.4 into subtopics with subheadings (e.g., "Preclinical Evidence", "Clinical Trials", "Shared Resistance").
- Although the explanation of resistance mechanisms is thorough, it occasionally repeats itself across pathways. For synthesis, a comparative table that summarizes the mechanisms (e.g., TME remodeling, PD-L1 overexpression, PTEN loss) would be very helpful.
- Table 1 is exhaustive, but it's difficult to interpret in a glance. Consider grouping trials by pathway (e.g., KRAS, FGFR) and adding highlights to denote trial phase or status.
Conclusion
The manuscript is ready for publication with a few small changes. Given that it addresses a crucial nexus in modern oncology, it is likely to be of interest to both clinical and scientific audiences. Focusing on synthesis and enhancing structure will further boost its impact and clarity.
Author Response
- Please define abbreviation (e.g., TME) at first usage.
TME has been defined on line 109
- Some languages are verbose: “A sandwich approach for short-term BRAF/MEKi…” could be more precise: “Short-term BRAF/MEKi followed by ICI (‘sandwich approach’) may preserve responsiveness before resistance emerges.”
We are grateful for the reviewer highlighting improvements in the language. We have reviewed the language in the manuscript and in addition updated that specific sentence as suggested
- The review paper is dense. For readability, please breaking down long paragraphs in sections 3.1-3.4 into subtopics with subheadings (e.g., “Preclinical Evidence", "Clinical Trials", "Shared Resistance").
Whilst we agree with the reviewer’s suggestion that breaking down paragraphs into preclinical evidence/clinical trials/shared resistance may improve readability, in practice it is very difficult to do for this subject. Within each sections, we feel that the breakdown of “1. Effect of oncogenic driver, 2. Combination therapy, 3. Shared resistance mechanisms 4. Summary” provides a clear and consistent format across
the review. Whilst some paragraphs and sections can be clearly split into preclinical and clinical trials, others require a mixture of evidence to discuss the particular topic.
- Although the explanation of resistance mechanisms is thorough, occasionally repeats itself across pathways. For synthesis, a comparative table that summarizes the mechanisms (e.g., TME remodelling, PD-L1 overexpression, PTEN loss) would be very helpful.
We have included a comparative table of the effect of combination therapy and resistance mechanisms to summarise shared features across pathways. (Table 2)
- Table 1 is exhaustive, but it's difficult to interpret in a glance. Consider grouping trials by pathway (e.g., KRAS, FGFR) and adding highlights to denote trial phase or status.
We have reformatted table 1 to make the distinction between different pathways clearer. (Table 1)
Reviewer 3 Report
Comments and Suggestions for Authors
Douglas et al. provide a comprehensive review examining the interplay between oncogenic signaling pathways and cancer immunotherapy efficacy. The manuscript thoroughly explores how aberrations in RAS/RAF/MAPK, FGFR, PI3K, and HER2 pathways modulate the tumor microenvironment and influence response to immune checkpoint inhibitors. While the authors present a detailed analysis of resistance mechanisms and combination strategies, I recommend addressing several areas to enhance the manuscript's clinical relevance and translational impact:
- The authors should elaborate on the logistical and technical barriers to implementing comprehensive tumor genomic profiling in routine clinical practice, particularly in resource-limited settings. Discussion of cost-effectiveness, turnaround time, and interpretation of complex molecular data would strengthen this section.
- The manuscript would benefit from a more systematic framework for determining optimal sequencing strategies across different molecular subtypes. While the BRAF-mutant melanoma paradigm is well-described, extrapolation of these principles to other oncogenic drivers requires further elaboration.
- A more nuanced discussion of toxicity management in combination regimens is warranted. Consideration of alternative dosing schedules, biomarker-guided patient selection, and preemptive supportive care strategies could address the concerning grade 3-4 adverse event profiles observed in pivotal trials.
- The authors should address the potential for developing predictive algorithms that integrate genomic, transcriptomic, and immune parameters to forecast resistance. Emerging computational approaches warrant discussion.
- Further critical appraisal of circulating tumor DNA monitoring is needed, specifically addressing sensitivity thresholds, concordance with tissue-based methods, and the actionability of detected alterations in clinical decision-making.
- For tumors harboring concurrent driver mutations, a hierarchical approach to pathway prioritization should be proposed. Discussion of competitive pathway crosstalk and compensatory signaling would enhance this section.
- The limitations of PD-L1 as a biomarker deserve more detailed examination, including assay variability, spatial heterogeneity, and temporal dynamics. Alternative or complementary biomarkers should be considered.
- A mechanistic model illustrating the potential synergistic interactions between intrinsic and extrinsic resistance mechanisms would significantly improve conceptual understanding of therapeutic resistance.
- Expanding the scope to include emerging driver alterations beyond the four major pathways discussed would enhance the manuscript's comprehensiveness and clinical utility.
Author Response
- The authors should elaborate on the logistical and technical barriers to implementing comprehensive tumor genomic profiling in routine clinical practice, particularly in resource-limited settings. Discussion of cost-effectiveness, turnaround time, and interpretation of complex molecular data would strengthen this section.
We have updated the discussion to highlight the challenges of comprehensive genomic profiling in clinical practice and resource poor settings. (Lines 702-709)
- The manuscript would benefit from a more systematic framework for determining optimal sequencing strategies across different molecular subtypes. While the BRAF-mutant melanoma paradigm is well-described, extrapolation of these principles to other oncogenic drivers requires further elaboration.
Outside of BRAF mutant melanoma, we feel it would be speculation to define the optimal sequencing strategy due to the lack of direct randomised trial data. However, we have proposed a framework of how to approach this clinical question which we have included as a second figure. (Lines 679 -687 plus figure 2)
- A more nuanced discussion of toxicity management in combination regimens is warranted. Consideration of alternative dosing schedules, biomarker-guided patient selection, and pre-emptive supportive care strategies could address the concerning grade 3-4 adverse event profiles observed in pivotal trials.
We feel that a detailed discussion of alternative dosing, pre-emptive supportive care strategies and biomarkers that predict toxicity is beyond the scope of this review as it is predominantly focused on mechanisms. However, we have referenced this in the discussion and highlighted that further work in this area is required. (Lines 665 – 670)
- The authors should address the potential for developing predictive algorithms that integrate genomic, transcriptomic, and immune parameters to forecast resistance. Emerging computational approaches warrant discussion.
We added in a future perspectives section to the discussion, highlighting new technologies, such as spatial technologies, that could be used to predict resistance. The use of these technologies is predominantly confined to preclinical research and not currently applicable to clinical practice. Therefore, we feel that including details of the computational approaches to integrate these technologies is beyond the scope of the review. (Line 688 onwards)
- Further critical appraisal of circulating tumor DNA monitoring is needed, specifically addressing sensitivity thresholds, concordance with tissue-based methods, and the actionability of detected alterations in clinical decision-making.
We have included details of the limitations and challenges of ctDNA analysis in the discussion addressing concordance with tissue samples and interpretation of low VAF alterations. (Lines 690-699)
- For tumors harboring concurrent driver mutations, a hierarchical approach to pathway prioritization should be proposed. Discussion of competitive pathway crosstalk and compensatory signaling would enhance this section.
The evidence of the effect of concurrent oncogenic drivers on immunotherapy response and development of shared resistance mechanisms is limited. We feel that to propose a hierarchy of pathway prioritisation would be speculation at this stage.
- The limitations of PD-L1 as a biomarker deserve more detailed examination, including assay variability, spatial heterogeneity, and temporal dynamics. Alternative or complementary biomarkers should be considered.
We feel that a detailed assessment of PD-L1 assays, variability and spatial heterogeneity is beyond the scope of the review.
- A mechanistic model illustrating the potential synergistic interactions between intrinsic and extrinsic resistance mechanisms would significantly improve conceptual understanding of therapeutic resistance.
We have updated section 2 to discuss the crosstalk between tumour cell intrinsic and extrinsic resistance mechanisms and how this may influence strategies to overcome immunotherapy resistance. (Lines 151-160)
- Expanding the scope to include emerging driver alterations beyond the four major pathways discussed would enhance the manuscript's comprehensiveness and clinical utility.
For this review we chose to focus on 4 common oncogenic drivers to allow a detailed analysis without making the article extremely lengthy. Analysis of other oncogenic pathways and targeted therapy would be extremely valuable but may be best served by a series of articles.